# Role of Inflammatory and Immune-Nutritional Prognostic Markers in Patients Undergoing Surgical Resection for Biliary Tract Cancers

**DOI:** 10.3390/cancers13143594

**Published:** 2021-07-18

**Authors:** Simone Conci, Tommaso Campagnaro, Elisa Danese, Ezio Lombardo, Giulia Isa, Alessandro Vitali, Ivan Marchitelli, Fabio Bagante, Corrado Pedrazzani, Mario De Bellis, Andrea Ciangherotti, Alfredo Guglielmi, Giuseppe Lippi, Andrea Ruzzenente

**Affiliations:** 1Division of General and Hepatobiliary Surgery, Department of Surgery, Dentistry, Pediatrics and Gynaecology, University of Verona, 37134 Verona, Italy; simone.conci@aovr.veneto.it (S.C.); tommaso.campagnaro@aovr.veneto.it (T.C.); ezio.lombardo@studenti.univr.it (E.L.); giulia.isa@studenti.univr.it (G.I.); alessandro.vitali@studenti.univr.it (A.V.); ivan.marchitelli@studenti.univr.it (I.M.); fabio.bagante@univr.it (F.B.); corrado.pedrazzani@univr.it (C.P.); mario.debellis_01@univr.it (M.D.B.); andrea.ciangherotti@gmail.com (A.C.); alfredo.guglielmi@univr.it (A.G.); 2Section of Clinical Biochemistry, Department of Neuroscience, Biomedicine and Movement Science, University of Verona, 37134 Verona, Italy; elisa.danese@univr.it (E.D.); giuseppe.lippi@univr.it (G.L.)

**Keywords:** biliary tract cancers, inflammatory and immune-nutritional markers, prognostic factors

## Abstract

**Simple Summary:**

Biliary tract cancers (BTCs) are a heterogeneous group of malignancies, which arise from the epithelial cells of the biliary tree, with a high rate of local invasion and metastatic spreading. Surgical resection remains the treatment which offers the best chance of long-term survival. However, new chemotherapy regimens and multimodal strategies have showed encouraging results, supporting the need for simple and readily available preoperative tools able to predict survival and guide the treatment strategy. Recently, the prognostic role of several nutritional and inflammatory indexes in growth, biological aggressiveness, and spread has been investigated in different types of cancers. Nevertheless, complete and conclusive results on BTCs are lacking. By identifying a preoperative immune and inflammatory prognostic index based on simple routine blood samples, we may have an additional element that is useful in guiding the treatment strategy by assigning selected patients to preoperative or postoperative treatments despite pathological results.

**Abstract:**

The relationship between immune-nutritional status and tumor growth; biological aggressiveness and survival, is still debated. Therefore, this study aimed to evaluate the prognostic performance of different inflammatory and immune-nutritional markers in patients who underwent surgery for biliary tract cancer (BTC). The prognostic role of the following inflammatory and immune-nutritional markers were investigated: Glasgow Prognostic Score (GPS), modified Glasgow Prognostic Score (mGPS), Prognostic Index (PI), Neutrophil to Lymphocyte ratio (NLR), Platelet to Lymphocyte ratio (PLR), Lymphocyte to Monocyte ratio (LMR), Prognostic Nutritional Index (PNI). A total of 282 patients undergoing surgery for BTC were included. According to Cox regression and ROC curves analysis for survival, LMR had the best prognostic performances, with hazard ratio (HR) of 1.656 (*p* = 0.005) and AUC of 0.652. Multivariable survival analysis identified the following independent prognostic factors: type of BTC (*p* = 0.002), T stage (*p* = 0.014), N stage (*p* < 0.001), histological grading (*p* = 0.045), and LMR (*p* = 0.025). Conversely, PNI was related to higher risk of severe morbidity (*p* < 0.001) and postoperative mortality (*p* = 0.005). In conclusion, LMR appears an independent prognostic factor of long-term survival, whilst PNI seems associated with worse short-term outcomes.

## 1. Introduction

Biliary tract cancers (BTCs) are the second most common hepatobiliary malignancy worldwide, representing about 3% of all gastrointestinal tumors [1]. They encompass a motley group of tumors, which arise from the epithelium of the biliary ducts. According to the site of origin, they are classified as intrahepatic cholangiocarcinoma (ICC) arising from intrahepatic bile ducts, perihilar cholangiocarcinoma (PCC) arising or involving the hepatic biliary confluence, distal cholangiocarcinoma (DCC) arising from the bile duct distal to the cystic duct origin, and gallbladder cancer (GBC) [2]. Regardless of their location, BTCs are very aggressive diseases, with a high rate of local invasion and metastatic spreading.

Surgical resection remains the treatment which offers the best chance of long-term survival, with several clinical and pathological prognostic factors (i.e., tumor size, lymph-node metastases, multiple nodules, vascular invasion and radicality of surgery) being now accepted as predictors of survival [3,4,5,6]. However, the encouraging results of new chemotherapy regimens and multimodal strategies support the need for simple and readily available preoperative tools, able to predict survival and guide the treatment strategy [7,8,9].

Great emphasis has recently been attributed to the role of nutritional and inflammatory status in growth, biological aggressiveness and spread of different types of cancers [10,11,12]. Previous studies investigated and often validated the prognostic role of several preoperative inflammatory and immune-nutritional markers in different malignancies [13,14,15,16,17,18,19,20]. However, although some data on ICC are available, complete and conclusive results on BTCs are lacking, so that the role of each proposed marker as predictor of survival remains controversial [21,22].

The aims of this study were to evaluate and compare the prognostic performances of different inflammatory and immune-nutritional markers on survival, and to investigate their possible relationship with postoperative short-term outcomes in patients who underwent surgical resection for BTC.

## 2. Materials and Methods

### 2.1. Patients

From September 2010 to December 2019, a total of 345 consecutive patients undergoing surgery for BTC in a referral center for HPB surgery were evaluated for the study. The clinical and pathological data of all the patients were prospectively collected. Only patients submitted to radical intent surgery, pathologically confirmed BTC, and with follow-up >6 months were enrolled in the current study. To this end, we had to exclude 40 patients who underwent explorative or palliative surgery, 7 patients with no diagnosis of BTC at the final pathology, and 16 patients with a follow-up <6 months. Therefore, a total number of 282 patients could be finally included.

Biliary tract cancers were classified according to the 3rd English edition of the Japanese classification of biliary tract cancers [2], as intrahepatic cholangiocarcinoma (ICC), perihilar cholangiocarcinoma (PCC), distal cholangiocarcinoma (DCC), and gallbladder cancer (GBC), and were staged according to the American Joint Committee against Cancer (AJCC) staging system 8th edition [23].

Serum white blood cell count (WBC), neutrophil, lymphocyte, monocyte count, platelet count (Plt), alanine aminotransferase (ALT), gamma-glutamyl transferase (GGT), alkaline phosphatase (ALP), total bilirubin, albumin (Alb), C-Reactive Protein (CRP), carcinoembryonic antigen (CEA), and carbohydrate antigen 19-9 (Ca 19-9) were measured within 2 weeks before surgery.

The inflammatory and nutritional prognostic markers were identified according to previous studies on different malignancies, as follows: Glasgow Prognostic Score (GPS), based on serum CRP and Alb [13]; modified Glasgow Prognostic Score (mGPS), based on serum CPR and Alb [14]; Prognostic Index (PI), based on serum CRP and WBC [15]; Neutrophil to Lymphocyte ratio (NLR) [16]; Platelet to Lymphocyte ratio (PLR) [17]; Lymphocyte to Monocyte ratio (LMR) [18]; Prognostic Nutritional Index (PNI), based on serum Alb and lymphocyte count [19].

Postoperative morbidity was recorded and classified according to Dindo-Clavien classification, where a grade ≥ 3 defined severe morbidity, whilst postoperative mortality was considered as in-hospital and/or 90-day mortality.

All patients underwent clinical, biochemical, and imaging (CT or MRI) follow-up every six months after surgery or in case of suspected recurrent disease.

Data collection and analysis were conducted according to institutional guidelines and conformed to the ethical standards of the World Medical Association (Declaration of Helsinki). The study was approved by the institutional review board and ethic committee of the University of Verona Hospital (Verona, Italy, No. 3260CESC). All patients included in the study subscribed a written informed consent.

### 2.2. Statistical Analysis

Continuous variables were reported as medians and interquartile ranges (IQR), while categorical variables were shown as numbers and percentages. Comparisons between groups were performed using the Student’s unpaired t-test for continuous or ordinal variables, while Chi square test or Fisher’s exact tests were adopted for categorical variables, when appropriate. Kruskal–Wallis and Mann–Whitney U test were used for comparing Ca 19-9 values between different groups.

The median follow-up period was 30.5 months (IQR 18.1–51.4) and 132 events occurred. Overall survival (OS) was defined as the interval between resection and death or of the last follow-up. The overall survival rates were calculated using the Kaplan–Meier method, and differences in survival rates between two groups were compared with log-rank test. Multivariable Cox proportion analysis was performed to assess the influence of factors on OS, which were found to have significant associations in univariate analysis.

To evaluate the discriminatory ability of each marker, receiver operating characteristics (ROC) curves were constructed at 12, 24, and 36 months after surgery, and the area under the curve (AUC) was calculated. The optimal cut-off for continuous variables was determined on ROC analysis and the relative sensitivity, specificity and Youden test was reported. For the comparison analysis (both survival and severe morbidity) between prognostic markers, category 1 and 2 of GPS, mGPS, and PI were grouped to obtain homogeneous comparison between dichotomized variables. Cox regression survival analysis, Akaike Information Criterion (AIC) calculated by multinomial logistic regression, and maximal metric statistics Kolmogorov–Smirnov (K–S) were used to compare the AUCs of inflammatory and immune-nutritional markers; the variables with the best performances (higher risk coefficients, lower AIC and higher max K–S) were included in multivariable analysis. Univariate and multivariable analysis for severe morbidity as well as postoperative mortality analysis were performed with logistic multinomial regression. Statistical analyses were performed using the SPSS statistical software package, version 28.0 (IBM SPSS Inc., Chicago, IL, USA), at a significance level of *p* less than 0.05.

## 3. Results

### 3.1. Clinical and Pathological Characteristics of the Study Population

The median age of the 282 patients with BTC included in the study was 69.5 (IQR 61.9 to 75.0) years. One hundred and twenty-nine patients (45.7%) had ICC, 94 (33.3%) PCC, 22 (7.8%) DCC, and 37 (13.1%) GBC. One hundred twenty-seven (45.0%) had jaundice at diagnosis and 113 (40.1%) required preoperative biliary drainage. Surgical resection required major hepatectomy in 164 patients (58.2%), pancreaticoduodenectomy in 23 (8.2%), and hepatoduodenectomy in 3 (1.1%). Biliary and vascular resection was performed in 168 (59.6%) and 28 (9.9%) patients, respectively. Resection (R0) was reached in 193 patients (68.4%). All patients underwent lymph-node dissection (100.0%) and the AJCC 8^th^ Ed. N stage was N0 in 153 patients (54.3%) and N1-2 in 129 (45.7%) patients. Severe morbidity (Dindo–Clavien ≥ 3) occurred in 62 patients (22.0%) and 5 patients died postoperatively (1.8%). Description of baseline clinical and pathological features are reported in Table 1. Comparison of clinical and pathological characteristics according to the type of BTC is summarized in Appendix A.

### 3.2. Inflammatory Based and Nutritional Prognostic Scores of the Study Population

The study population was stratified according to the prognostic scores as follows: Glasgow Prognostic Score (GPS) 0, 192 (68.1%) patients, GPS 1, 63 (22.3%) patients, and GPS 2, 27 (9.6%) patients; modified Glasgow Prognostic Score (mGPS) 0, 214 (75.9%), mGPS 1 41 (14.5%) patients, and mGPS 2, 27 (9.6%) patients; Prognostic Index (PI) 0, 200 (70.9) patients, PI 1, 58 (20.6%), and PI 2, 24 (8.5%). The median (IQR) values of NLR, PLR, LMR, and PNI were 3.09 (2.04–4.05), 160.4 (117.3–230.3), 3.34 (2.39–4.62), and 49.4 (44.3–53.6), respectively. According to ROC curves analysis, the optimal cut-off values for continuous inflammatory and immune-nutritional markers were 3.13 for NLR, 178.2 for PLR, 3.47 for LMR, and 48.6 for PNI, respectively. Details on inflammatory and immune-nutritional markers, and their distribution in the study population are reported in Table 2. A comparison of inflammatory and immune-nutritional markers according to the type of BTC is reported in Appendix A.

### 3.3. Prognostic Value of Inflammatory and Immune-Nutritional Prognostic Markers

In univariate analysis, GPS (hazard ratio [HR], 1.623; 95% C.I., 1.217–2.153, *p* < 0.001), mGPS (HR, 1.537; 95% C.I., 1.198–2.019, *p* = 0.004), PI (HR, 1.559; 95% C.I., 1.235–2.105, *p* = 0.001), and PNI (HR, 1.450; 95% C.I., 1.030–2.041, *p* = 0.033) were significantly associated with survival; based on HR values, LMR (HR, 1.656; 95% C.I., 1.167–2.351, *p* = 0.005) was found to be the best predictor of overall survival (Table 3). The results of ROC curves analysis for survival at 12, 24, and 36 months are summarized in Table 4.

At 12 months, GPS had the best performances (AUC, 0.706; sensitivity, 0.714; specificity, 0.722; Youden Index, 0.436). At 24 and 36 months, LMR had the best performances (AUC, 0.623 and 0.652; sensitivity, 0.671 and 0.633; specificity, 0.548 and 0.633; Youden Index, 0.219 and 0.266, respectively). The cumulative survival curves calculated for the different inflammatory and immune-nutritional markers are shown in Figure 1.

### 3.4. Prognostic Factors of Overall Survival

The 5-years OS rate of the whole study population was 35.8%, with median survival of 46.4 months (41.2–51.6 months). The 5-years OS rate progressively decreased among the type of BTC: 42.7% in patients with ICC, 37.2% in patients with DCC, 31.8% in patients with PCC, and 20.7% in patients with GBC (*p* < 0.001), respectively (Appendix A). The following prognostic factors have been identified at the univariate analysis: AJCC 8th T stage (*p* < 0.001), AJCC 8th N stage (*p* < 0.001), histologic grading (*p* = 0.002), radicality of surgery (*p* = 0.006), macrovascular invasion (*p* < 0.001), and LMR (*p* = 0.004). Multivariable analysis recognized the following independent prognostic factors for OS: type of BTC (overall *p* = 0.002; ICC: ref; PCC: HR 1.512; 95% C.I., 1.016–2.295, *p* = 0.048; DCC: HR 0.935; 95% C.I., 0.461–1.895, *p* = 0.935; GBC: HR 2.698; 95% C.I., 1.580–4.606, *p* < 0.001), AJCC 8th T stage (HR, 1.515; 95% C.I., 1.076–2.352, *p* = 0.014), AJCC 8th N stage (HR 2.431; 95% C.I., 1.613–3.664, *p* < 0.001), histologic grading (HR, 1.494; 95% C.I., 1.008–2.213, *p* = 0.045), and LMR (HR, 1.378; 95% C.I., 1.046–2.007, *p* = 0.025). Univariate and multivariate analysis of prognostic factors for OS are summarized in Table 5. Overall survival curves according to the main clinical and pathological prognostic factors are shown in Appendix A.

### 3.5. Short Term Outcomes According to the Inflammatory and Nutritional Prognostic Markers

Interestingly, several prognostic markers seem to be related to severe morbidity (Dindo–Clavien ≥ 3): GPS (*p* < 0.001), mGPS (*p* = 0.008;), PI (*p* = 0.013), NLR (*p* = 0.004), PLR (*p* = 0.003), and PNI (*p* < 0.001). Univariate analysis of the different prognostic markers for severe morbidity are summarized in Table 6.

Clinical factors related to postoperative severe morbidity in univariate analysis were: Age >70 years (*p* = 0.044), presence of PBD (*p* < 0.001), major hepatectomy (*p* = 0.001), biliary resection (*p* = 0.001). The multivariable analysis revealed that presence of PBD (OR, 2.504; *p* = 0.031) and PNI (OR, 3.109; *p* = 0.003) were independent predictors of severe morbidity.

Univariate and multivariable analysis with other relevant variables for severe morbidity were performed and are summarized in Table 7. Postoperative mortality occurred in only 5 patients (1.8%). Among inflammatory and immune-nutritional prognostic markers PNI (OR, 8.547; 95% C.I., 1.876–13.789, *p* = 0.005) seem to be related to a high risk of postoperative mortality in univariate analysis (Appendix A). However, these association were not confirmed in multivariable analysis.

## 4. Discussion

The surgical outcomes in BTC patients are strongly linked to the presence of known prognostic factors, such as local tumor extension, presence of lymph-node or distant metastases, radicality of surgery, as well as involvement of vascular or peri-hepatic structures [3,4,5,6]. More focus has recently been turned toward the inflammatory and immune-nutritional background, in which the tumor develops, and several studies have highlighted the impact of chronic inflammation in development and biological behaviour of different types of neoplasms. Considering BTC, elevated serum levels of pro-inflammatory cytokines, such as tumor necrosis factor (TNF)-α and interleukin (IL)-6, seems related to colangiocarcinogenesis, promoting peritumoral lymph-angiogenesis [24,25]. The host immune response seems to play a key role in tumor spreading and progression: the number of circulating activated neutrophils increases in the presence of a tumor and the resulting inflammatory state promotes tumor proliferation and neo-angiogenesis, mediated by Vascular Endothelial Growth Factor (VEGF) [26]; furthermore, neutrophilia can inhibit the adaptive immune response, mediated by lymphocytes [27]. A study by Zhou et al. on a series of 60 ICC patients demonstrated that tumor cells gain the direct ability to mobilize and attract neutrophils to the tumor site by overexpression and secretion of chemokines, such as CXCL5 [28]. According to the results of a recent study published by Tanaka et al. on 154 surgically resected BTCs, an inverse correlation was found between circulating neutrophils and lymphocytes, and NLR seems to predict the prognosis [29]. Platelets are also involved in the inflammatory response against the tumor, whereby platelet adhesion and aggregation lead to the formation and release of granules which contain proteases, growth factors (e.g., TGFβ, VEGF, PDGF), and cytokines that seem to support tumor progression [30].

Nutritional status is closely linked with the proper functioning of the host’s immune system; in fact, malnutrition can lead to immune function disorder, as well as neutrophil, macrophage, and lymphocyte dysfunction [31]. Several studies reported the detrimental impact of malnutrition on the short- and long-term outcomes of different solid tumors, increasing the risk of postoperative complications and tumor progression [32,33,34]. Immune-nutritional markers, such as PNI, have shown a prognostic role also in patients with cholangiocarcinoma in both curative or palliative treatment strategies [35,36]. Okuno et al. evaluated the prognostic value of mGPS, NLR, PLR, and PNI in a surgical series of 534 PCC, concluding that, in this subgroup of BTC and among the prognostic markers tested, only mGPS seemed to stratify the long-term survival. Unfortunately, the study reported no data on postoperative course and its relationship with the immune-nutritional or inflammatory markers [36].

Kitano et al. compared the prognostic performances of NLR and PLR on 120 patients who underwent surgery for extrahepatic cholangiocarcinoma (including PCC and DCC) and demonstrated that high PLR (rather than high NLR) seems to be associated with both recurrence-free and overall survival [37]. Unlike this evidence, a similar study on a multi-institutional series of 991 resected ICC showed that NLR seems to have an important and independent prognostic impact in this specific subtype of BTC [38].

Lin et al. were the first to compare and validate the prognostic value of GPS, mGPS, PI, NLR, PLR, LMR, and PNI in a series of 200 patients with ICC in 2019 [21]. Although such study demonstrated a significant prognostic value of all these scores, LMR was found to be the best predictive test in multivariable analysis (HR, 2.082; *p* = 0.007). The prognostic relevance of LMR has also been demonstrated in other liver malignancies, such as hepatocellular carcinoma.

The current study seems consistent with results of this latter study, although differences have emerged, particularly in the threshold values of the prognostic markers. Contrary to previous studies, our cohort of patients included all the types of BTC and some differences in the levels of the prognostic markers among the types of BTC have been identified (Appendix A). This could explain the mild discrepancies between our thresholds and the others reported in the literature. Considering BTC cumulatively, and comparing several inflammatory and immune-nutritional markers, LMR (HR, 1.656; *p* = 0.005) was confirmed as the strongest OS predictor. Moreover, LMR maintained its prognostic role in multivariable analysis, after multiple comparison with the leading clinical and pathological prognostic factors.

A reasonable explanation to support our findings could be the important role that monocytes/macrophages play in influencing tumor growth, angiogenesis, and metastatic spreading. The pro-inflammatory cytokines, growth factors, and chemokines released by monocytes/macrophages seem to be involved in reducing activated CD4+ and CD8+ T lymphocytes, which are responsible for the specific anti-neoplastic immune response [39,40,41].

Finally, we investigated the relationship between inflammatory and immune-nutritional markers and short-term postoperative outcomes. Malnutrition, inflammatory status, and cachexia caused by the presence of a neoplasm are typically mirrored by lower protein synthesis, so that the predictably low serum albumin concentration could be associated with high postoperative mortality, as reported for several cancers [42]. In our study, despite several markers showing an association with severe morbidity and postoperative mortality, PNI was found to be the strongest predictor of both severe morbidity and postoperative mortality in patients undergoing surgery for biliary tract cancer (Table 6 and Appendix A). Nonetheless, the predictive value of PNI was not retained in multivariable analysis, and this is perhaps attributable to the limited number of postoperative deaths registered, which may have impaired the power of our statistical analysis.

In recent years, emphasis has been placed on enhancing the effectiveness of the immune system and nutritional status in an effort to improve the results of major abdominal surgery. However, several trials failed to demonstrate a direct efficacy of a wide and routine administration of immune-nutrition support [43,44,45,46]. However, the extensive use of immune-nutritional and inflammatory markers to identify high-risk patients could lead to a tailored prehabilitation program including immune-nutritional support when required.

A limitation of the current study could be considered to be its retrospective nature, resulting in the potential presence of selection biases in patients undergoing surgery. The relatively long period in which data has been collected may have also impacted our findings, since the surgical techniques, perioperative management, and chemotherapy regimens have significantly improved over time. Moreover, the design of the study and the proposal of new thresholds for continuous markers may limit the reproducibility of our results. Therefore, external validation with additional prospective studies would be needed to confirm our findings in order to improve reliability, reproducibility, and clinical usefulness of such inflammatory and immune-nutritional markers.

## 5. Conclusions

The prognosis of patients with BTC seems to be directly influenced by inflammatory and immune-nutritional status. In particular, LMR, a pure inflammatory marker, appears an independent prognostic factor of long-term survival, whilst PNI, an immune-nutritional marker, seems associated with worse short-term outcomes.

## Figures and Tables

**Figure 1 cancers-13-03594-f001:**
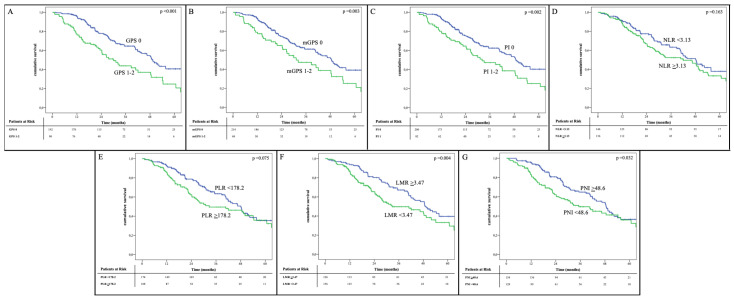
Survival curves according to the inflammatory and nutritional prognostic scores: (**A**) Glasgow Prognostic Score (GPS); (**B**) modified Glasgow Prognostic Score (mGPS); (**C**) Prognostic Index (PI); (**D**) Neutrophil to Lymphocyte ratio (NLR); (**E**) Platelet to Lymphocyte ratio (PLR); (**F**) Lymphocyte to Monocyte ratio (LMR); (**G**) Prognostic Nutritional Index (PNI).

**Table 1 cancers-13-03594-t001:** Clinical and pathological characteristics of the study population.

Characteristics		Patients n 282
Age, years, median (IQR)		69.5 (61.9–75.0)
Gender, male, *n* (%)		167 (59.2)
CA 19.9, U/mL, median (IQR)		209.0 (33.3–1418.3)
Type of BTC, *n* (%)	ICC	129 (45.7) 94 (33.3) 22 (7.8) 37 (13.1)
	PCC	
	DCC	
	GBC	
Jaundice, *n* (%)		127 (45.0)
Preoperative Biliary Drainage (PBD), *n* (%)		113 (40.1)
Preoperative Chemotherapy, *n* (%)		12 (4.3)
Portal Vein Embolization (PVE), *n* (%)		9 (3.2)
Tumor size, mm, median (IQR)		50 (30–60)
Type of Surgery, *n* (%)	Atypical Liver Resection	5 (1.8)
	Segmentectomy	13 (4.6)
	Bisegmentectomy	60 (21.3)
	Left Hepatectomy	79 (28.0)
	Right Hepatectomy	53 (18.8)
	Mesohepatectomy	7 (2.5)
	Left Trisectionectomy	10 (3.5)
	Right Trisectionectomy	7 (2.5)
	Other Major Hepatectomy	7 (2.5)
	Pancreaticoduodenectomy	23 (8.2)
	Common Bile Duct Resection	14 (5.0)
	Cholecystectomy	1 (0.3)
	Hepatoduodenectomy	3 (1.1)
Extent of Hepatectomy, *n* (%)	Minor	80 (28.4)
	Major	164 (58.2)
	n/a	38 (13.4)
Biliary Resection, *n* (%)		168 (59.6)
Vascular resection, *n* (%)		28 (9.9)
AJCC 8th Ed. T Stage, *n* (%)	T1-2	129 (45.7)
	T3-4	153 (54.3)
AJCC 8th Ed. N Stage, *n* (%)	N0	153 (54.3)
	N1-2	129 (45.7)
Histologic Grading	G1-2	196 (69.5)
	G3-4	73 (25.9)
	n/a	13 (4.6)
Macrovascular invasion, *n* (%)		74 (26.2)
Microvascular invasion, *n* (%)		199 (70.6)
Radicality of Surgery, *n* (%)	R0	193 (68.4)
	R1	89 (31.6)
Severe Morbidity (Clavien–Dindo ≥ 3), *n* (%)		62 (22.0)
Postoperative mortality, *n* (%)		5 (1.8)
Hospital stay, days, median (IQR)		12 (7–19)
Postoperative chemotherapy		181 (64.2)

IQR, interquartile range; AJCC, American Joint Committee against Cancer.

**Table 2 cancers-13-03594-t002:** Inflammatory and immune-nutritional prognostic markers and its distribution among the study population.

Inflammatory or Immune-Nutritional Marker	Score	Patients n 282 *n* (%)
**Glasgow Prognostic Score (GPS)**		
CRP ≤ 10 mg/L and albumin ≥ 35 g/L	0	192 (68.1)
CRP ≤ 10 mg/L and albumin < 35 g/L	1	22 (7.8) 41 (14.5)
CRP > 10 mg/L and albumin ≥ 35 g/L	1	
CRP > 10 mg/L and albumin < 35 g/L	2	27 (9.6)
**Modified Glasgow Prognostic Score (mGPS)**		
CRP ≤ 10 mg/L and albumin ≥ 35 g/L	0	192 (68.1) 22 (7.8)
CRP ≤ 10 mg/L and albumin < 35 g/L	0	
CRP > 10 mg/L and albumin ≥ 35 g/L	1	41 (14.5)
CRP > 10 mg/L and albumin < 35 g/L	2	27 (9.6)
**Prognostic Index (PI)**		
CRP ≤ 10 mg/L and WBC ≤ 11 × 10^9^/L	0	200 (70.9)
CRP ≤ 10 mg/L and WBC > 11 × 10^9^/L	1	50 (17.7)
CRP > 10 mg/L and WBC ≤ 11 × 10^9^/L	1	8 (2.8)
CRP > 10 mg/L and WBC > 11 × 10^9^/L	2	24 (8.5)
**Neutrophil to lymphocyte ratio (NLR),** median (IQR)		3.09 (2.04–4.05)
NLR < 3.13 *	0	146 (51.8)
NLR ≥ 3.13 *	1	136 (48.2)
**Platelet to lymphocyte ratio (PLR),** median (IQR)		160.4 (117.3–230.3)
PLR < 178.2 *	0	174 (61.7)
PLR ≥ 178.2 *	1	108 (38.3)
**Lymphocyte to monocyte ratio (LMR),** median (IQR)		3.34 (2.39–4.62)
LMR < 3.47 *	1	156 (55.3)
LMR ≥ 3.47 *	0	126 (44.7)
**Prognostic Nutritional Index (PNI),** median (IQR)		49.4 (44.3–53.6)
Albumin (g/L) + 5 × total lymphocyte count < 48.6 *	1	128 (45.4)
Albumin (g/l) + 5 × total lymphocyte count ≥ 48.6 *	0	154 (54.6)

GPS, Glasgow Prognostic Score; mGPS, modified Glasgow Prognostic Score; PI, Prognostic Index; NLR, neutrophil to lymphocyte ratio; PLR, platelet to lymphocyte ratio; LMR, lymphocyte to monocyte ratio; PNI, prognostic nutritional index; CRP, C-reactive protein; WBC, white blood cells; IQR, Interquartile range; * The optimal cut-off of NLR, PLR, LMR e PNI was determined by ROC curves analysis.

**Table 3 cancers-13-03594-t003:** Univariate analysis and cox logistic regression analysis for overall survival of the inflammatory based and nutritional prognostic markers.

Prognostic Marker		5-Year OS (%)	HR	95% C.I.	*p* Values
GPS	0	40.7	1.628	1.217–2.153	<0.001
	1–2	24.7			
mGPS	0	39.2	1.537	1.198–2.019	0.004
	1–2	25.1			
PI	0	40.3	1.559	1.235–2.105	0.001
	1–2	25.3			
NLR	<3.13	37.9	1.274	0.906–1.793	0.164
	≥3.13	33.1			
PLR	<178.2	35.4	1.366	0.967–1.928	0.077
	≥178.2	35.8			
LMR	≥3.47	39.6	1.656	1.167–2.351	0.005
	<3.47	33.2			
PNI	≥48.6	36.5	1.450	1.030–2.041	0.033
	<48.6	35.9			

GPS, Glasgow Prognostic Score; mGPS, modified Glasgow Prognostic Score; PI, Prognostic Index; NLR, neutrophil to lymphocyte ratio; PLR, platelet to lymphocyte ratio; LMR, lymphocyte to monocyte ratio; PNI, prognostic nutritional index.

**Table 4 cancers-13-03594-t004:** ROC curves analysis of the inflammatory-based and nutritional prognostic factors.

Prognostic Marker	Overall Survival AUC *	Sensitivity	Specificity	Youden Index	AIC ^§^	Max K–S ^#^
12 months						
GPS	0.706	0.714	0.722	0.436	104.650	0.436
mGPS	0.642	0.500	0.789	0.289	99.128	0.289
PI	0.651	0.571	0.738	0.309	98.870	0.310
NLR	0.558	0.536	0.527	0.063	101.090	0.159
PLR	0.603	0.536	0.629	0.165	99.846	0.164
LMR	0.590	0.714	0.477	0.191	99.310	0.191
PNI	0.664	0.750	0.582	0.332	100.732	0.332
24 months						
GPS	0.604	0.471	0.742	0.213	129.729	0.213
mGPS	0.555	0.314	0.794	0.108	128.051	0.108
PI	0.561	0.386	0.742	0.128	127.751	0.128
NLR	0.577	0.571	0.555	0.126	129.477	0.126
PLR	0.603	0.600	0.600	0.200	130.853	0.193
LMR	0.623	0.671	0.548	0.219	127.393	0.235
PNI	0.608	0.629	0.606	0.235	130.117	0.220
36 months						
GPS	0.602	0.439	0.776	0.215	131.604	0.214
mGPS	0.555	0.306	0.806	0.112	127.562	0.112
PI	0.559	0.378	0.745	0.123	127.022	0.122
NLR	0.576	0.551	0.541	0.092	129.026	0.092
PLR	0.563	0.480	0.643	0.123	128.589	0.122
LMR	0.652	0.633	0.633	0.266	126.807	0.265
PNI	0.611	0.643	0.571	0.214	128.579	0.184

GPS, Glasgow Prognostic Score; mGPS, modified Glasgow Prognostic Score; PI, Prognostic Index; NLR, neutrophil to lymphocyte ratio; PLR, platelet to lymphocyte ratio; LMR, lymphocyte to monocyte ratio; PNI, prognostic nutritional index. AUC, area under curve, * The AUC values reported were cathegorized/dichotomized based on optimal cut-off identified by ROC curves analysis; ^§^ Akaike Information Criterion; ^#^ maximal Statistics Kolmogorov–Smirnov.

**Table 5 cancers-13-03594-t005:** Univariable and multivariable analysis for overall survival.

Characteristics		Univariable Analysis			Multivariable Analysis		
		5-Years OS (%)	Median OS (Months)	*p* Values	HR	95% C.I.	*p* Values
Age, years	<70	41.9	49.4	0.081			
	≥70	27.4	38.7				
Gender	M	30.3	43.9	0.120			
	F	43.3	51.3				
Type of BTC	ICC	42.7	50.1	<0.001	ref		0.002
	PCC	31.8	41.8		1.512	1.016–2.295	0.048
	DCC	37.2	31.2		0.935	0.461–1.895	0.935
	GBC	20.7	18.6		2.698	1.580–4.606	<0.001
Preoperative ChT	No	35.5	43.1	0.793			
	Yes	38.1	46.5				
AJCC 8th T Stage	T1–2	48.1	55.7	<0.001	1.515	1.076–2.352	0.014
	T3–4	23.8	30.4				
AJCC 8th N Stage	N0	53.4	64.9	<0.001	2.431	1.613–3.664	<0.001
	N1–N2	12.8	26.3				
Histologic grading	G1–2	36.9	48.5	0.002	1.494	1.008–2.213	0.045
	G3–4	28.5	26.9				
Radicality	R0	38.2	48.8	0.006	1.108	0.746–1.647	0.611
	R1	32.1	32.8				
Macrovascular invasion	No	39.0	48.8	<0.001	1.335	0.894–1.991	0.157
	Yes	27.4	26.6				
Postoperative ChT	No	35.0	43.1	0.654			
	Yes	37.3	48.1				
LMR	≥3.47	39.6	48.8	0.004	1.378	1.046–2.007	0.025
	<3.47	33.2	32.8				

BTC, biliary tract cancer; ChT, chemotherapy; AJCC, American Joint Committee against Cancer; PBD, preoperative biliary drainage; LMR, lymphocyte to monocyte ratio.

**Table 6 cancers-13-03594-t006:** Severe morbidity according to the different inflammatory and immune-nutritional markers.

Prognostic Marker		Univariate Analysis		Logistic Regression		
		*n* (%)	*p* Values	OR	95% C.I.	*p* Values
GPS	0	27/192 (14.1)	<0.001	3.889	2.161–6.998	<0.001
	1–2	35/90 (38.9)				
mGPS	0	39/214 (18.2)	0.007	2.293	1.245–4.223	0.008
	1–2	23/68 (33.8)				
PI	0	36/200 (18.0)	0.010	2.115	1.174–3.810	0.013
	1–2	26/82 (31.7)				
NLR	<3.13	22/146 (15.1)	0.003	2.348	1.309–4.213	0.004
	≥3.13	40/136 (29.4)				
PLR	<178.2	28/174 (16.1)	0.002	2.396	1.351–4.249	0.003
	≥178.2	34/108 (31.5)				
LMR	≥3.47	22/126 (17.5)	0.066	1.630	0.909–2.922	0.101
	<3.47	40/156 (25.6)				
PNI	≥48.6	16/154 (10.4)	<0.001	4.838	2.574–9.095	<0.001
	<48.6	46/128 (35.9)				

GPS, Glasgow Prognostic Score; mGPS, modified Glasgow Prognostic Score; PI, Prognostic Index; NLR, neutrophil to lymphocyte ratio; PLR, platelet to lymphocyte ratio; LMR, lymphocyte to monocyte ratio; PNI, prognostic nutritional index.

**Table 7 cancers-13-03594-t007:** Univariable and multivariable analysis for severe morbidity.

Prognostic Marker/		Univariate Analysis		Multivariable Analysis		
Characteristics		*n* (%)	*p* Values	OR	95% C.I.	*p* Values
Age, years	<70	27/152 (17.8)	0.044	1.785	0.927–3.438	0.083
	≥70	35/130 (26.9)				
Gender	M	31/115 (27.0)	0.064			
	F	31/167 (18.6)				
Type of BTC	ICC	23/129 (17.8)	0.089			
	PCC	29/94 (30.9)				
	DCC	4/22 (18.2)				
	GBC	6/37 (16.2)				
PBD	No	24/169 (14.2)	<0.001	2.504	1.090–5.753	0.031
	Yes	38/113 (33.6)				
Major	No	9/80 (11.3)	0.001	1.980	0.762–5.140	0.161
Hepatectomy	Yes	47/164 (28.7)				
Biliary	No	14/114 (12.3)	<0.001	0.859	0.313–2.537	0.768
Resection	Yes	48/168 (28.6)				
Vascular	No	55/254 (21.7)	0.421			
Resection	Yes	7/28 (25.0)				
PNI	≥48.6	16/154 (10.4)	<0.001	3.109	1.475–6.554	0.003
	<48.6	46/128 (35.9)				

BTC, biliary tract cancer; PBD, preoperative biliary drainage; PNI, prognostic nutritional index.

## Data Availability

The data presented in the study are available on request from the corresponding author. The data are not publicly available due to ethical restrictions.

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
