# Peer review of "Role of Inflammatory and Immune-Nutritional Prognostic Markers in Patients Undergoing Surgical Resection for Biliary Tract Cancers"

_cancers, 2021, doi:10.3390/cancers13143594_

Round 1
Reviewer 1 Report
Dear Authors
The paper is well written and developed, and performed a thorough analysis searching the possible prognostic role of several inflammatory and nutritional scores. The Authors conclude that LMR appears an independent prognostic factor of long-term survival, whilst PNI, an immune-nutritional marker, seems associated with worse short-term outcomes.
Even if the topic is interesting, there are in the Literature different and sometimes controversial results with regarding the scores the Authors analyzed. The Authors briefly report (in the Discussion section) some other papers reporting results on inflammatory factors/scores and hepatic surgery. I believe that they need to better discuss previously reported papers dealing with nutritional scores and outcomes in hepatic oncologic surgery.
Furthermore, I would add a comment regarding the potential clinical relevance of assessing such scores, in terms of therapeutic impact.
Finally, I would modify the Table 2, by specifying how many patients for each subcategorization of the scores are present
Kind regards
Author Response
Reviewer #1:
The paper is well written and developed, and performed a thorough analysis searching the possible prognostic role of several inflammatory and nutritional scores. The Authors conclude that LMR appears an independent prognostic factor of long-term survival, whilst PNI, an immune-nutritional marker, seems associated with worse short-term outcomes.
Q1. Even if the topic is interesting, there are in the Literature different and sometimes controversial results with regarding the scores the Authors analyzed. The Authors briefly report (in the Discussion section) some other papers reporting results on inflammatory factors/scores and hepatic surgery. I believe that they need to better discuss previously reported papers dealing with nutritional scores and outcomes in hepatic oncologic surgery.
A1. Thanks for the suggestion. We add a specific paragraph in the discussion
The text has been modified as follows and references updated:
Discussion section:
“...Nutritional status is closely linked with the proper functioning of the host's immune system, in fact, malnutrition can lead to immune function disorder, and neutrophil, macrophage, and lymphocyte dysfunction [31]. Several studies reported the detrimental impact of malnutrition on the short- and long-term outcomes of different solid tumors, increasing the risk of postoperative complications and tumor progression [32-34]. Immune-nutritional markers, such as PNI, have shown a prognostic role also in patients with cholangiocarcinoma in both curative or palliative treatment strategies [35-36]. Okuno et al. evaluated the prognostic value of mGPS, NLR, PLR and PNI in a surgical series of 534 PCC concluding that, in this subgroup of BTC and among the prognostic markers tested, only mGPS seems to stratify the long-term survival. Unfortunately, the study reported no data on postoperative course and its relationship with the immune-nutritional or inflammatory markers [36]. ”
- Jayarajan, S.; Daly, J.M. The relationships of nutrients, routes of delivery, and immunocompetence. Surg Clin North Am. 2011 Aug;91(4):737-53. doi: 10.1016/j.suc.2011.04.004. PMID: 21787965.
- Schwegler, I.; von Holzen, A.; Gutzwiller, J.P.; Schlumpf, R.; Mühlebach, S.; Stanga, Z . Nutritional risk is a clinical predictor of postoperative mortality and morbidity in surgery for colorectal cancer. Br J Surg. 2010;97:92–7. doi: 10.1002/bjs.6805. PMID: 20013933.
- Kanda, M.; Fujii, T.; Kodera, Y.; Nagai, S.; Takeda, S.; Nakao, A. Nutritional predictors of postoperative outcome in pancreatic cancer. Br J Surg. 2011 Feb;98(2):268-74. doi: 10.1002/bjs.7305. PMID: 20960457.
- Zhang, W.; Ye, B.; Liang, W.; Ren, Y. Preoperative prognostic nutritional index is a powerful predictor of prognosis in patients with stage III ovarian cancer. Sci Rep. 2017 Aug 25;7(1):9548. doi: 10.1038/s41598-017-10328-8.
- Cui, P.; Pang, Q.; Wang, Y.; Qian, Z.; Hu, X.; Wang, W.; Li, Z.; Zhou, L.; Man, Z.; Yang, S.; Jin, H.; Liu, H. Nutritional prognostic scores in patients with hilar cholangiocarcinoma treated by percutaneous transhepatic biliary stenting combined with 125I seed intracavitary irradiation: A retrospective observational study. Medicine (Baltimore). 2018 Jun;97(22):e11000. doi: 10.1097/MD.0000000000011000. PMID: 29851859.
- Okuno, M.; Ebata, T.; Yokoyama, Y.; Igami, T.; Sugawara, G.; Mizuno, T.; Yamaguchi, J.; Nagino, M. Evaluation of inflammation-based prognostic scores in patients undergoing hepatobiliary resection for perihilar cholangiocarcinoma. J Gastroenterol. 2016 Feb;51(2):153-61. doi: 10.1007/s00535-015-1103-y.
Q2. Furthermore, I would add a comment regarding the potential clinical relevance of assessing such scores, in terms of therapeutic impact.
A2. Thank you. We add a specific paragraph in the discussion
The text has been modified as follows and references updated:
Discussion section:
“...In recent years, emphasis has been placed on enhancing the effectiveness of the immune system and nutritional status in an effort to improve the results of major abdominal surgery. However, several trials failed to demonstrated a direct efficacy of widely and routinely administration of immune-nutrition support [43-46]. However, the extensive use of immune-nutritional and inflammatory markers to identified high risk patients could lead to a tailored prehabilitation program including immune-nutritional support when required...”
- Klek, S.; Kulig, J.; Sierzega, M.; Szybinski, P.; Szczepanek, K.; Kubisz, A.; Kowalczyk, T.; Gach, T.; Pach, R.; Szczepanik, A.M. The impact of immunostimulating nutrition on infectious complications after upper gastrointestinal surgery: a prospective, randomized, clinical trial. Ann Surg. 2008 Aug;248(2):212-20. doi: 10.1097/SLA.0b013e318180a3c1. PMID: 18650630.
- Lobo, D.N.; Williams, R.N.; Welch, N.T.; Aloysius, M.M.; Nunes, Q.M.; Padmanabhan, J.; Crowe, J.R.; Iftikhar, S.Y.; Parsons, S.L.; Neal, K.R.; Allison, S.P.; Rowlands, B.J. Early postoperative jejunostomy feeding with an immune modulating diet in patients undergoing resectional surgery for upper gastrointestinal cancer: a prospective, randomized, controlled, double-blind study. Clin Nutr. 2006 Oct;25(5):716-26. doi: 10.1016/j.clnu.2006.04.007. Epub 2006 Jun 13. PMID: 16777271.
- Burden, S.; Todd, C.; Hill, J.; Lal, S. Pre-operative nutrition support in patients undergoing gastrointestinal surgery. Cochrane Database Syst Rev. 2012 Nov 14;11:CD008879. doi: 10.1002/14651858.CD008879.pub2. PMID: 23152265.
- Hughes, M.J.; Hackney, R.J.; Lamb, P.J.; Wigmore, S.J.; Christopher Deans, D.A.; Skipworth, R.J.E. Prehabilitation Before Major Abdominal Surgery: A Systematic Review and Meta-analysis. World J Surg. 2019 Jul;43(7):1661-1668. doi: 10.1007/s00268-019-04950-y. PMID: 30788536.
Q3. Finally, I would modify the Table 2, by specifying how many patients for each subcategorization of the scores are present
A3. Thanks. Table 2 has been modified according to the reviewer’s suggestion.
Reviewer 2 Report
Dear colleages,
this is a comprehensive analysis of the prospective value of several nutritional and inflammatory markers based on a large cohort of a single tertiary referral center.
However, the cutoff-values of NLR, PLR, LMR, PNI are generated by ROC-curve analysis within your own data set. I would appreciate a comment on the cutoff-values of NLR, PLR, LMR, PNI used by other authors like Lin et al.
Is there a universal cutoff-value and would there be a similar discrimination in your data set if you would use a different cut-off-value?
Thank you for a short comment on that point.
Author Response
Reviewer #2:
This is a comprehensive analysis of the prospective value of several nutritional and inflammatory markers based on a large cohort of a single tertiary referral center.
Q1. However, the cutoff-values of NLR, PLR, LMR, PNI are generated by ROC-curve analysis within your own data set. I would appreciate a comment on the cutoff-values of NLR, PLR, LMR, PNI used by other authors like Lin et al.
A1. Thank you for the comment. The study by Lin et al. included only patients with intrahepatic cholangiocarcinoma (ICC), as reported in supplementary table 2, there are differences in the values of prognostic markers according to the type of BTC. Consequently, it is not surprising that the cut-off values identified by Lin et al. differs from the cut-off values of the current study that include a more heterogeneous and complex group of tumors.
The text has been modified as follows:
Discussion section:
“...The current study appears to be consistent with the results of this latest study, although differences have emerged in particular in the threshold values of the prognostic markers. Conversely to previous studies, our cohort of patients included all the type of BTC and some differences in levels of the prognostic markers among the types of BTC have been identified (supplementary table 2). This could explain the mild discrepancies between our thresholds and the other reported in literature.”
Q2. Is there a universal cutoff-value and would there be a similar discrimination in your data set if you would use a different cut-off-value?
Thank you for a short comment on that point.
A2. Unfortunately, it is very difficult to compare the studies published on this topic at the time. Every study has different design, different study population, compares different prognostic markers and use different cut-off values. Larger and homogeneous series are mandatory to improve reliability, reproducibility and clinical usefulness of the prognostic markers. Probably further studies could be specifically designed to identify a universal cut-off of a single prognostic marker without the comparison of different scores and subsequently it could be verified in external validation cohorts. Thanks for the inspiration for a new study.
The text has been modified as follows:
Discussion section:
“...Therefore, external validation with additional prospective studies would be needed to confirm our findings in order to improve reliability, reproducibility and clinical usefulness of such inflammatory and immune-nutritional markers.”
Reviewer 3 Report
The authors present a series of patients resected for BTC, with the aim of studying the prognostic value of different scores related to inflammation, and to compare them.
The number are relatively high for these rare tumors, however different primary locations were pooled, which might be both a strength and a limitation. These scores were previously confirmed in multiple series (some much larger than the present one) as prognostic, the novelty might have been the comparison of them (albeit it was also previously done in ICC); however, the limitations of the methodology do not allow to draw definitive conclusions.
Main comments:
- There is no validation cohort that might help to confirm the validity of the conclusions.
- As frequently done with these scores, the authors choose a new “best” threshold, which is always varying between studies… the conclusions with the threshold chosen will thus be limited to their series, as others might find different results with their own best threshold in their own series.
- Some variables are studied with 3 categories vs 2 categories; that will influence the results (for example, values of HRn which was the criterion to select LMR for multivariate analysis and draw the main conclusion, also values of AUC). The continuous scores were not studied as such but as dichotomized variables; results might have been different if used as continuous or categorize with 3 categories…
- The authors might consider, partly in response to previous comment, to group together categories 1 and 2 of the scores with 3 categories, as number of patients with high score is low and overall differences between 1 and 2 do not appear significant.
- Only 1 HR were provided for scores with 3 categories, as if difference for 1 vs 0 was similar to 2 vs 1, which is obviously not the case when looking at the curves.
- The authors should clarify if the AUC value provided for the continuous scores on Table 4 are those of the continuous score or the dichotomized score.
- AUCs could be compared statistically.
- The authors should use other measures of the value of a score: Akaike Information Criterion, Harrell’s c statistics, to better address their objective of selecting the best score.
- The authors should provide median follow-up and number of events.
- Analysis of the morbidity parameters was done according to thresholds used for prognostic values, which is not justifiable.
- I don’t understand why Table 6 and 7 are not replaced by a single analysis of both clinical and non-clinical parameters. Could the authors explain.
- Could the authors please present data on missing data?
Author Response
Reviewer #3:
The authors present a series of patients resected for BTC, with the aim of studying the prognostic value of different scores related to inflammation, and to compare them.
The number are relatively high for these rare tumors, however different primary locations were pooled, which might be both a strength and a limitation. These scores were previously confirmed in multiple series (some much larger than the present one) as prognostic, the novelty might have been the comparison of them (albeit it was also previously done in ICC); however, the limitations of the methodology do not allow to draw definitive conclusions.
Main comments:
Q1. There is no validation cohort that might help to confirm the validity of the conclusions.
A1. Thanks for the comment. We discussed the opportunity to divide the study population in training and validation cohort, however the decrease of sample size could weak the statistical power and significance of the analysis. We believe that further study and external validation could better confirm our results.
Q2. As frequently done with these scores, the authors choose a new “best” threshold, which is always varying between studies… the conclusions with the threshold chosen will thus be limited to their series, as others might find different results with their own best threshold in their own series.
A2. We agree with the reviewer that is a limitation of our study as other published in literature. This study was designed to investigate the importance of different immune and nutritional markers, and to compare their value in this particular type of tumors. With this aim we selected the best threshold to have the best available performance of each factor in the current series. A comment was added in the discussion.
Text has been modified as follows:
Discussion section
“...Moreover, the design of the study and the proposal of new thresholds for continuous markers may limit the reproducibility of our results...”
Q3. Some variables are studied with 3 categories vs 2 categories; that will influence the results (for example, values of HRn which was the criterion to select LMR for multivariate analysis and draw the main conclusion, also values of AUC). The continuous scores were not studied as such but as dichotomized variables; results might have been different if used as continuous or categorize with 3 categories…
Q4. The authors might consider, partly in response to previous comment, to group together categories 1 and 2 of the scores with 3 categories, as number of patients with high score is low and overall differences between 1 and 2 do not appear significant.
Q5. Only 1 HR were provided for scores with 3 categories, as if difference for 1 vs 0 was similar to 2 vs 1, which is obviously not the case when looking at the curves.
A3-4-5. We agree with the reviewer and this is the main challenge we had to overcome to compare different markers both continuous and categorized. We re-performed the analysis according to the reviewer suggestion.
The text has been modified as follows:
Methods section:
“...For the comparison analysis (both survival and severe morbidity) between prognostic markers, category 1 and 2 of GPS, mGPS and PI were grouped to obtain homogeneous comparison between dichotomized variables...”
Figure 1 A-C were redesigned
Table 3:
Table 3. Univariate analysis and cox logistic regression analysis for overall survival of the inflammatory based and nutritional prognostic markers.
Prognostic Marker |
5-year OS (%) |
HR |
95% C.I. |
p values |
GPS |
0 |
40.7 |
1.628 |
1.217-2.153 |
<0.001 |
|
1-2 |
24.7 |
mGPS |
0 |
39.2 |
1.537 |
1.198-2.019 |
0.004 |
|
1-2 |
25.1 |
|||
PI |
0 |
40.3 |
1.559 |
1.235-2.105 |
0.001 |
|
1-2 |
25.3 |
|||
NLR |
< 3.13 |
37.9 |
1.274 |
0.906-1.793 |
0.164 |
|
≥ 3.13 |
33.1 |
|||
PLR |
< 178.2 |
35.4 |
1.366 |
0.967-1.928 |
0.077 |
|
≥ 178.2 |
35.8 |
|||
LMR |
≥ 3.47 |
39.6 |
1.656 |
1.167-2.351 |
0.005 |
|
< 3.47 |
33.2 |
|||
PNI |
≥ 48.6 |
36.5 |
1.450 |
1.030-2.041 |
0.033 |
|
< 48.6 |
35.9 |
Table 6: reported below
Q6. The authors should clarify if the AUC value provided for the continuous scores on Table 4 are those of the continuous score or the dichotomized score.
A6. The AUC values for continuous prognostic markers (NLR, PLR, LMR, and PNI) reported in Table 4 are of the continuous score, and the sensitivity, specificity and Youden test reported are relative to the optimal cut-off identified.
The text has been modified as follows:
Methods section:
“...and the relative sensitivity, specificity and Youden test reported...”
Table 4. Legend
“...AUC, area under curve, *The AUC values reported were from categorized (GPS, mGPS and PI) or continuous (NLR, PLR, LMR and PNI) variables based on the nature of the prognostic markers and the sensitivity, specificity and Youden test based on optimal cut-off identified by ROC curves analysis”
Q7. AUCs could be compared statistically. The authors should use other measures of the value of a score: Akaike Information Criterion, Harrell’s c statistics, to better address their objective of selecting the best score.
A7. We really appreciate the comment and suggestion. We performed the comparison analysis between AUCs with Akaike Infromation Criterion test and with Statistics Kolmogorov-Smirnov. The results confirmed the hypothesis that LMR seems to be the best predictor of long-term survival and were added in the Table 4.
The text has been modified as follows:
Methods section:
-“...Cox regression survival analysis, Akaike Information Criterion (AIC) calculated by multinomial logistic regression, and maximal metric statistics Kolmogorov-Smirnov were used to compare the AUCs of inflammatory and immune-nutritional markers; the variable with the best performances (higher risk coefficients, lower AIC and higher max K-S)were included in multivariable analysis.
-“...Statistical analyses were performed using the SPSS statistical software package, version 28.0 (IBM SPSS Inc., Chicago, IL, USA), at a significance level of p less than 0.05...”
Table 4:
Table 4. ROC curves analysis of the inflammatory-based and nutritional prognostic factors.
Prognostic Marker |
Overall Survival AUC |
Sensitivity |
Specificity |
Youden Index |
AIC§ |
Max K-S# |
12 months |
|
|
|
|
|
|
GPS |
0.706 |
0.714 |
0.722 |
0.436 |
104.650 |
0.436 |
mGPS |
0.642 |
0.500 |
0.789 |
0.289 |
99.128 |
0.289 |
PI |
0.651 |
0.571 |
0.738 |
0.309 |
98.870 |
0.310 |
NLR |
0.558 |
0.536 |
0.527 |
0.063 |
101.090 |
0.159 |
PLR |
0.603 |
0.536 |
0.629 |
0.165 |
99.846 |
0.164 |
LMR |
0.590 |
0.714 |
0.477 |
0.191 |
99.310 |
0.191 |
PNI |
0.664 |
0.750 |
0.582 |
0.332 |
100.732 |
0.332 |
24 months |
|
|
|
|
|
|
GPS |
0.604 |
0.471 |
0.742 |
0.213 |
129.729 |
0.213 |
mGPS |
0.555 |
0.314 |
0.794 |
0.108 |
128.051 |
0.108 |
PI |
0.561 |
0.386 |
0.742 |
0.128 |
127.751 |
0.128 |
NLR |
0.577 |
0.571 |
0.555 |
0.126 |
129.477 |
0.126 |
PLR |
0.603 |
0.600 |
0.600 |
0.200 |
130.853 |
0.193 |
LMR |
0.623 |
0.671 |
0.548 |
0.219 |
127.393 |
0.235 |
PNI |
0.608 |
0.629 |
0.606 |
0.235 |
130.117 |
0.220 |
36 months |
|
|
|
|
|
|
GPS |
0.602 |
0.439 |
0.776 |
0.215 |
131.604 |
0.214 |
mGPS |
0.555 |
0.306 |
0.806 |
0.112 |
127.562 |
0.112 |
PI |
0.559 |
0.378 |
0.745 |
0.123 |
127.022 |
0.122 |
NLR |
0.576 |
0.551 |
0.541 |
0.092 |
129.026 |
0.092 |
PLR |
0.563 |
0.480 |
0.643 |
0.123 |
128.589 |
0.122 |
LMR |
0.652 |
0.633 |
0.633 |
0.266 |
126.807 |
0.265 |
PNI |
0.611 |
0.643 |
0.571 |
0.214 |
128.579 |
0.184 |
- Akaike Information Criterion; # maximal Statistics Kolmogorov-Smirnov
Q9. The authors should provide median follow-up and number of events.
A9. The median follow up period was 30.5 months (IQR 18.1-51.4) with 132 events during the follow up period
The text has been modified as follows:
Methods section:
“...The median follow up period was 30.5 months (IQR 18.1-51.4) and 132 events occurred...”
Q10. Analysis of the morbidity parameters was done according to thresholds used for prognostic values, which is not justifiable.
A10. Thanks for the comment. We performed a specific ROC curves analysis on Severe Morbidity and identified the optimal cut-off values for severe morbidity (see Table X). As you can see they are not too different from the previous cut-off identified by OS analysis. Moreover, we performed univariate and logistic regression analysis for Severe Morbidity (see Table XX). Not surprisingly the performances of the different markers seems improve by the use of specific cut-off values, however PNI remain the best predictor of Severe Morbidity. According to Q2 and the comments of Reviewer #2, we believe that including other cut-off values in the manuscript could be confusing and make the manuscript difficult to interpret, given that the meaning of the paper remain the same. However, we could modify the manuscript according to further revision or editor decision.
Table X. ROC curves analysis for Severe Morbidityof the inflammatory-based and nutritional prognostic factors.
Prognostic Marker (cut-off value*) |
Severe Morbidity AUC |
Sensitivity |
Specificity |
Youden Index |
AIC§ |
Max K-S# |
GPS |
0.649 |
0.565 |
0.750 |
0.315 |
145.107 |
0.315 |
mGPS |
0.581 |
0.371 |
0.795 |
0.166 |
138.639 |
0.166 |
PI |
0.581 |
0.419 |
0.745 |
0.164 |
138.771 |
0.165 |
NLR (3.15) |
0.635 |
0.645 |
0.582 |
0.227 |
140.856 |
0.244 |
PLR (168.5) |
0.641 |
0.645 |
0.618 |
0.263 |
141.541 |
0.263 |
LMR (3.21) |
0.592 |
0.597 |
0.559 |
0.156 |
140.368 |
0.195 |
PNI (49.3) |
0.746 |
0.806 |
0.582 |
0.591 |
135.136 |
0.397 |
*for continuous markers; §Akaike Information Criterion; # maximal Statistics Kolmogorov-Smirnov
Table XX. Severe morbidity according to the different inflammatory and immune-nutritional markers.
Prognostic Marker |
Univariate Analysis |
Logistic regression |
|||||
|
n (%) |
P values |
OR |
95% C.I. |
P values |
||
GPS |
0 |
27/192 (14.1) |
< 0.001 |
3.889 |
2.161-6.998 |
<0.001 |
|
|
1-2 |
35/90 (38.9) |
|
|
|
||
mGPS |
0 |
39/214 (18.2) |
0.007 |
2.293 |
1.245-4.223 |
0.008 |
|
|
1-2 |
23/68 (33.8) |
|
|
|
||
PI |
0 |
36/200 (18.0) |
0.010 |
2.115 |
1.174-3.810 |
0.013 |
|
|
1-2 |
26/82 (31.7) |
|
|
|
||
NLR |
< 3.15 |
22/150 (14.7) |
0.001 |
2.530 |
1.409-4.541 |
0.002 |
|
|
≥ 3.15 |
40/132 (30.3) |
|
|
|
||
PLR |
< 168.5 |
22/158 (13.9) |
< 0.001 |
2.944 |
1.637-5.295 |
<0.001 |
|
|
≥ 168.5 |
40/124 (32.3) |
|
|
|
||
LMR |
≥ 3.21 |
25/147 (17.0) |
0.025 |
1.842 |
1.039-3.267 |
0.037 |
|
|
< 3.21 |
37/135 (27.4) |
|
|
|
||
PNI |
≥ 49.3 |
15/146 (10.3) |
< 0.001 |
4.612 |
2.431-8.751 |
<0.001 |
|
|
< 49.3 |
47/136 (34.6) |
|
|
|
||
Q11. I don’t understand why Table 6 and 7 are not replaced by a single analysis of both clinical and non-clinical parameters. Could the authors explain.
A11. Table 6 report the univariate analysis and the logistic regression of Severe Morbidity and Postoperative Mortality according to the different prognostic markers. PNI (the one with the best performances) was included in the multivariable analysis on Severe Morbidity with the other clinical factors and reported in Table 7. Probably due to the low postoperative mortality rate of the study population (5/282, 1.8%) the results of a similar analysis on Mortality was not significant and was not reported. However, we believe that a single table including both the analysis (identifications of best prognostic marker and multivariable analysis) could be confusing and preferred keep it separated. We decided to remove the analysis on postoperative mortality (reported in a new Supplementary Table 3).
The text has been modified as follows:
Table 6
Table 6. Severe morbidity according to the different inflammatory and immune-nutritional markers.
Prognostic Marker |
Univariate Analysis |
Logistic regression |
|||||
|
n (%) |
P values |
OR |
95% C.I. |
P values |
||
GPS |
0 |
27/192 (14.1) |
< 0.001 |
3.889 |
2.161-6.998 |
<0.001 |
|
|
1-2 |
35/90 (38.9) |
|
|
|
||
mGPS |
0 |
39/214 (18.2) |
0.007 |
2.293 |
1.245-4.223 |
0.008 |
|
|
1-2 |
23/68 (33.8) |
|
|
|
||
PI |
0 |
36/200 (18.0) |
0.010 |
2.115 |
1.174-3.810 |
0.013 |
|
|
1-2 |
26/82 (31.7) |
|
|
|
||
NLR |
< 3.13 |
22/146 (15.1) |
0.003 |
2.348 |
1.309-4.213 |
0.004 |
|
|
≥ 3.13 |
40/136 (29.4) |
|
|
|
||
PLR |
< 178.2 |
28/174 (16.1) |
0.002 |
2.396 |
1.351-4.249 |
0.003 |
|
|
≥ 178.2 |
34/108 (31.5) |
|
|
|
||
LMR |
≥ 3.47 |
22/126 (17.5) |
0.066 |
1.630 |
0.909-2.922 |
0.101 |
|
|
< 3.47 |
40/156 (25.6) |
|
|
|
||
PNI |
≥ 48.6 |
16/154 (10.4) |
< 0.001 |
4.838 |
2.574-9.095 |
<0.001 |
|
|
< 48.6 |
46/128 (35.9) |
|
|
|
||
Table 7
Table 7. Univariable and multivariable analysis for severe morbidity.
Prognostic Marker/ |
Univariate Analysis |
Multivariable Analysis |
||||
Characteristics |
n (%) |
P values |
OR |
95% C.I. |
P values |
|
Age, years |
< 70 |
27/152 (17.8) |
0.044 |
1.785 |
0.927-3.438 |
0.083 |
|
> 70 |
35/130 (26.9) |
|
|
|
|
Gender |
M |
31/115 (27.0) |
0.064 |
|
|
|
|
F |
31/167 (18.6) |
|
|
|
|
Type of BTC |
ICC |
23/129 (17.8) |
0.089 |
|
|
|
|
PCC |
29/94 (30.9) |
|
|
|
|
|
DCC |
4/22 (18.2) |
|
|
|
|
|
GBC |
6/37 (16.2) |
|
|
|
|
PBD |
No |
24/169 (14.2) |
<0.001 |
2.504 |
1.090-5.753 |
0.031 |
|
Yes |
38/113 (33.6) |
|
|
|
|
Major |
No |
9/80 (11.3) |
0.001 |
1.980 |
0.762-5.140 |
0.161 |
Hepatectomy |
Yes |
47/164 (28.7) |
|
|
|
|
Biliary |
No |
14/114 (12.3) |
<0.001 |
0.859 |
0.313-2.537 |
0.768 |
Resection |
Yes |
48/168 (28.6) |
|
|
|
|
Vascular |
No |
55/254 (21.7) |
0.421 |
|
|
|
Resection |
Yes |
7/28 (25.0) |
|
|
|
|
PNI |
≥ 48.6 |
16/154 (10.4) |
< 0.001 |
3.109 |
1.475-6.554 |
0.003 |
|
< 48.6 |
46/128 (35.9) |
|
|
|
|
Supplementary Table 3
Supplementary Table 3. Mortality according to the different inflammatory and immune-nutritional markers.
Prognostic Marker |
Univariate Analysis |
Logistic regression |
|||||
|
n (%) |
P values |
OR |
95% C.I. |
P values |
||
GPS |
0 |
1/192 (0.5) |
0.037 |
8.884 |
0.978-20.661 |
0.052 |
|
|
1-2 |
4/90 (4.4) |
|
|
|
||
mGPS |
0 |
2/214 (0.9) |
0.093 |
4.892 |
0.800-19.913 |
0.086 |
|
|
1-2 |
3/68 (4.4) |
|
|
|
||
PI |
0 |
2/200 (1.0) |
0.149 |
3.759 |
0.616-22.929 |
0.151 |
|
|
1-2 |
3/82 (3.7) |
|
|
|
||
NLR |
< 3.13 |
3/146 (2.1) |
0.533
|
0.711 |
0.117-4.324 |
0.712 |
|
|
≥ 3.13 |
2/136 (1.5) |
|
|
|
||
PLR |
< 178.2 |
3/174 (1.7) |
0.635
|
1.075 |
0.177-6.542 |
0.937 |
|
|
≥ 178.2 |
2/108 (1.9) |
|
|
|
||
LMR |
≥ 3.47 |
3/126 (2.4) |
0.400
|
0.532 |
0.088-3.237 |
0.494 |
|
|
< 3.47 |
2/156 (1.3) |
|
|
|
||
PNI |
≥ 48.6 |
1/154 (0.6) |
0.018
|
8.547 |
1.876-13.789 |
0.005 |
|
|
< 48.6 |
4/128 (3.1) |
|
|
|
||
Q12. Could the authors please present data on missing data?
A12. I suppose the Reviewer refers to “n/a” reported in Table 1. Thirty-eight patients did not underwent hepatic resection (major nor minor) but 23 pancreaticoduodenectomy, 14 common bile duct resection and 1 cholecystectomy. In 13 specimen (4.6%) data of histologic grading are not available in the local pathology archives.
Round 2
Reviewer 3 Report
The authors revised thoroughly their manuscript and answered most comments.
Some remaining points about my previous questions:
Q3 to Q5: very similar HR between GPS and LMR; I think no conclusion could be drawn about the relative value of both scores.
Q9: please add as a limitation that results at 36 months are associated with uncertainties as median follow-up is only 30 months.
Q12: sorry for the lack of clarity, I was asking whether some baseline characteristics (clinical or lab values) might have been missing. Do you really have CA 19.9 for all patients, for example? Same question for all variables included in all scores.
However, I feel that the limitations of the work (definition of optimal - for this dataset- thresholds, no validation cohort, no clear and significant differences in results) do not allow to support the main conclusions that one score is better than another.